# Inclusion Goals: What Sex Education for LGBTQIA+ Adolescents?

**DOI:** 10.3390/children11080966

**Published:** 2024-08-10

**Authors:** Stefano Eleuteri, Marta Girardi, Rossella Spadola, Elisabetta Todaro

**Affiliations:** Institute of Clinical Sexology, 00198 Rome, Italy; martagirardi@hotmail.it (M.G.); roxyspa77@gmail.com (R.S.); elisabettatodaro@gmail.com (E.T.)

**Keywords:** inclusive sex education, LGBTQIA+ youths, sexual rights, parent-adolescent sex communication, school programs, sex educator professional program

## Abstract

According to the World Health Organization and the World Association for Sexual Health (WAS) Declaration of Sexual Rights, sex education aims to provide children and adolescents with comprehensive knowledge, skills, attitudes, and values, promoting equality and nondiscrimination while upholding freedom of thought and expression. However, current school curricula often focus on dichotomous sexual identity and hetero-cis-normative prevention strategies, neglecting the needs of gender minority (GM) and sexual minority (SM) youths. In family settings, discussions typically revolve around sexual risk reduction and basic contraception, omitting relational aspects and components of sexual identity such as orientations and gender expressions. This discrepancy highlights a gap between the official goals of sex education and its practical implementation, reflecting a cultural deficit in familial sex education. This study reviews the scientific literature on sexual health promotion interventions from 2015 to 2024 to identify inclusive approaches that enhance the participation of all youths, not just hetero-cis-normative individuals. The hypothesis is that most interventions primarily serve heterosexual and cisgender youths, indicating a need for more inclusive strategies to achieve better sexual health and educational outcomes. The study also suggests expanding curricula to align with recognized guidelines and the diverse needs of youth.

## 1. Introduction

According to the World Health Organization [1,2] and the World Association for Sexual Health (WAS) Declaration of Sexual Rights [3], sex education aims to provide children and adolescents with comprehensive knowledge, skills, values, and attitudes promoting equality and nondiscrimination while upholding freedom of thought and expression.

The WHO-Regionalbüro für Europa und BZgASalute Standards [4] state that sexuality education means “learning relative to the cognitive, emotional, social, relational, and physical aspects of sexuality” and enhancing “the empowerment of children and young people by providing them with positive information, skills, and values for understanding their own sexuality and enjoying it, having safe and rewarding relationships, and behaving responsibly with respect to their own and others’ sexual health and well-being. (…)” (p. 17).

At an international level, governments and education policy makers are reviewing the best practices in the delivery of relationship and sexuality education (RSE) [5]. Even if there are some changes to the most recent guidelines, like the New Zealand ones [6], modifications are mostly connected to a greater recognition of gender diversity rather than the active inclusion of diverse sexualities. These other issues could be mitigated by reimagining sexuality education in a way that could include explicit reference to sexual practices but also the possibility to adopt a ‘norm-critical’ approach to sexuality education, like the use of innovative strategies such as the careful choice of language to avoid unnecessary gendering and the adoption of an overt ‘norm-critical’ approach to avoid assumptions [7]. In other jurisdictions, inclusivity is more take into consideration, but is focused on cultural diversity rather than LGBTQIA+ perspectives [8].

For several years now, what were previously distinguished as different branches of sex education (“Sex Education” and “Comprehensive Sex Education”), have converged into the single representation of holistic sex education, promoting an integrated (between informational and emotional aspects), sex-positive, and inclusive perspective, accessible according to an intersectional view.

Adolescence can be considered as one of the most critical transitions in the life of the individual, during which facing a process of growth and change, characterized by different levels of maturation (physical, sexual, psycho-emotional, etc.), pushes adolescents toward the acquisition of dimensions such as social independence, identity development, and relational and emotional competencies.

As ascertained by leading scholars in Developmental Psychology and Developmental Psychosexology, this process may be more complex in cases where the individual is faced with the development of a non-heterosexual or non-cisgender sexual identity [9]. In this scenario, an additional element of difficulty may arise precisely from being differently included in sexual health promotion programs [10].

Talking about non-heterosexual and non-cisgender identities refers to movements to deconstruct the hetero-cis-normative framework, which defines “natural” and “normal” identities and behavior based on a complementarity between male and female. According to the hetero-cis-normative view, gender is thought of as corresponding to biological sex and as a dichotomous variable, and male and female are found to be mutually attractive, for both romantic and sexual and sentimental purposes. In sexuality education programs, one of the main limitations represented by this view in the context of sex education is the rigid prescription of stereotypes regarding gender identity and gender expression, as well as limiting knowledge and exploration of all legitimate components of sexual identity. LGBTQIA+ is the abbreviation for “Lesbian, Gay, Bisexual, Transgender, Queer, Intersex, and Asexual” people. The additional “+” stands for all of the other identities not encompassed in the shorter acronym. It is an umbrella term that is often used to refer to the community of non-heterosexual and non-cisgender identities as a whole.

According to evidence from the scientific literature, sex education is more effective on heterosexual youth than on LGBTQIA+ youth [11]. This finding can be explained by the insufficient inclusion of Sexual Minorities (SMs) and Gender Minorities (GMs) within programs. Therefore, the current most widely used sex education is poorly able to meet needs, identities, and subjectivities, entailing the risk of being an information and growth tool only for a narrow part of the target audience for which it is designed. Thus, it may be interesting to have some risk-related data. In general, SM and GM youths experience more negative outcomes, relative to sexual health, than heterosexual peers; for example, homosexual male youths have an extremely higher risk of HIV infections [10].

As evident from Figure 1, SM youth who are recipients of hetero-cis-normative sex education, in addition to being exposed to greater sexual risks, also reveal worse psychological health status, reporting higher rates of attempted suicide, substance abuse, depression, and anxiety, compared to their heterosexual cisgender counterparts [12,13,14,15].

What Figure 1 intends to show is the multidimensional (sexual, psychological, and relational) complex of negative factors that hetero-cis-normative culture entails for the health of young people belonging to SMs and GMs. Indeed, in societies and cultures strongly anchored in the hetero-cis-normative framework, the absence of a constructivist view of sexual identity and the lack of promotion of free self-knowledge can lead young people to incur in nonconscious, unwanted, and non-respectful experiences.

Effects such as these can be discussed by correlating them with the stressors that, following the Minority Stress model [16], are specific to SMs and GMs, originating phenomena such as internalized homo/transphobia, stigma, bullying, discrimination, and violence [15].

LGBTQIA+ youths who face bullying and bias in educational settings are more likely to suffer academically, displaying lower grade point averages, higher rates of absenteeism, a sense of alienation from their school environment, and diminished self-esteem, compared to their peers who do not encounter such victimization and discrimination at school [17].

The objective of this article is to review the published scientific literature in the field of sex education in order to highlight the presence or absence of inclusive programs and methodologies of work in the adolescent target audience; a further aim is to advance a critical discussion of the effects of the currently most popular programs and proposals for the use of more inclusive intervention methodologies [18,19]. The use of inclusive sex education aims to decrease school-based victimization that characterizes the experiences of SM and GM youth, the impact of sexually transmitted diseases (of which LGBTQIA+ youth are found to be one of the most at-risk groups), and the risk of experiencing violent relationships [20].

LGBTQIA+ students who received inclusive sexual health education showed the following outcomes: diminished levels of victimization, higher feelings of safety at school, less safety-related school absences, increased academic performance, and more feelings of connection with peers [21].

Evidence-based programs have also shown having an impact on sexual behavior. However, these programs are often not inclusive of LGBTQIA+ identities, thus limiting program effectiveness in the general population.

Programs that have taken more inclusive measures still have gaps, especially in the relationship aspect.

Recent research [22] has showed that digital spaces can be positively correlated with social support, self-esteem, diminished sense of loneliness, increasing identity socialization, expression of emotions, and development of positive interpersonal relationships.

It is important that practitioners and families consider digital tools as valuable allies, and turn their attention to a more inclusive and participatory promotion of sex education using both these tools and traditional ones.

In considering the use of a systemic perspective (which intends to involve all significant adults in the educational intervention on sexuality), it is important to consider educational action as a process of community empowerment and as a political action. Through educational processes on sexuality, it is possible to promote more inclusive and respectful communication skills, and to enhance dialogue within family units and peers. One example among all may be the recent debate on the use of inclusive language, which has raised (in some countries such as Italy) trends of opposition to the possibility of dealing with the topic of gender identity within school sex education curricula, especially in elementary school. At the same time, it has raised the debate on the topic, leading many parents, teachers, and educators to confront and activate processes of questioning some of the stereotypes present in the culture of reference.

## 2. Materials and Methods

This narrative review is based on the results produced by a search of several databases, such as Academic Search Index, CINAHL, Complementary Index, EBCO, Scopus, Medline, MLA International Bibliography, PubMed, Science Direct, and Springer Nature. The selected articles refer to publications dated from 2015 through 2024. The keywords used for the search are “LGBTQIA+ comics”, “LGBTQIA+ inclusive sex education”, “LGBTQIA+ narratives”, “LGBTQIA+ youth”, “sexual communication”, “LGBTQIA+ sexual education”, and “LGBTQIA+ students” AND “parent-adolescent sex communication”. Based on the results obtained and the purpose of this paper, the most relevant studies were selected. Subsequently, the references and citations in the articles were used to expand the documentation. The data from the studies and the authors’ positions were analyzed to frame the current “state of the art” in order to identify critical issues in current inclusive programs and promote new educational intervention strategies. To be included in this review, a quantitative study had to meet all the following inclusion criteria:should have been published in peer-reviewed journals during or since 2015;should have focused on LGBTQIA+ population;should have been published in English and Spanish language.

Papers were excluded if:inclusive criteria were not present.

Since this is a narrative review, no other guidelines were followed during the search and the selection of the studies.

In Figure 2, a flow chart of the study selection procedure is provided.

## 3. Results

### 3.1. The Effects of Sex Education Targeting Heterosexuals

Most of the evidence on inclusive policies comes from the United States; it is generally focused on the impact of school-level policies rather than state-level policies [23,24]. In a study conducted by Evans, heterosexual young adult women reported receiving more information about sex before becoming sexually active than bisexual or lesbian young adult women [25]. Another research suggests that transgender identities are completely excluded from the contents addressed in sexuality education [26]. Regarding sexual risk, overall, SM and GM youths experience more negative sexual health outcomes than their heterosexual peers. It is found in the literature that gay and bisexual women run a higher risk of sexually transmitted infections (STIs) and unwanted pregnancies [27]. In addition, among transgender youth, those younger than 13 years old are three times more prone to experience sexual experimentation [28].

### 3.2. Suggestions for Improving Inclusive Sex Education

Research highlights the importance of dedicated youth work for LGBTQ+ youth, providing them with a safe, nonjudgmental, and welcoming space to explore their identities with professionals who can serve as positive LGBTQIA+ role models and to have the opportunity to network with peers what they can share the journey of knowing their identities [29,30].

Current sex education programs have taken measures to be more inclusive, but they still have gaps, especially in the relationship aspect, and students rarely actually receive sex health education inclusive of LGBTQIA+ population [31]. In the educational curriculum reported in the Chicago City Agency Newsletter, starting in 2nd grade (7–8 years old), students learn the distinction between gender and sex. In the 5th grade (ages 10–11), the specific topic of LGBTQIA+ identities is introduced, which is then brought up again in the 13–16 age range. In addition, issues such as the risk of STIs are addressed through various types of sexual stimulation (such as oral, anal, manual, and vaginal intercourse), and issues like insemination, adoption and surrogacy are considered for heterosexual and LGBTQIA+ families. Finally, guidelines have been adopted to support transgender and gender-nonconforming youths by ensuring they have access to facilities that legitimize them and allow them to affirm their gender identity in all school settings [32].

### 3.3. Perceptions of the Youth Population

In 2018, Hobaica and Know studied SM and GM students’ perceptions of sexuality education programs, finding that participants perceived their sexuality education to be permeated with normative elements.

Absence of visibility of all different sexual orientations, shame, and a lack of information on safe sexual practices emerged. This results in a feeling of unpreparedness for sexuality, which leads LGBTQIA+ youth to use other resources for self-education [33].

As summarized in Figure 3, a study of the sex and relationship education (SRE) experiences of young bisexual and queer women in Tasmania [34] found that most had received information on menstruation, puberty, contraceptive methods, information on preventing STIs, and reproduction. However, only a small percentage reported addressing issues related to gender, sexuality, and relationships. Complaints from participants in SRE programs included an excessive focus on biological aspects and a reproduction of gender inequalities through a focus on risk, personal responsibility, ethics, and punitive self-regulation, as well as an exclusive focus on risk in heterosexual relationships [34]. The points highlighted by this study, as shown in Figure 3, need to be taken into account when rethinking sex education programs from a perspective that can cross-cut the interests of all youth targets; the question educators can ask is: Is it possible to make the bio-psycho-social elements more inclusive? An example may be sexual anatomy: is there a way to be able to represent everyone’s experience and perception?

An additional study conducted in the United States [35] collected information on the primary and secondary school sex education experiences of transgender youth (AMAB, AFAB, nonbinary) across a wide range of programs. These reports severely lack information on various identities, such as the complete absence of information on queer identities.

The experience of invisibility that characterizes the experiences of young people belonging to SMs and GMs with respect to sexuality education means that it is perceived as “uncomfortable” and not very conducive to the possibility of formulating the questions and elaborating on them in the growth of identity awareness and understanding. Pregnancy, abortion, STIs, and slut-shaming are the topics mainly addressed to promote fear and abstinence-oriented behaviors [36]. In addition, anatomical and biological aspects are often addressed in sex education programs with a division of classes according to a binary gender criterion (males and females) resulting in the exclusion of individuals belonging to GMs. Complaints from participants about sexual education programs concerned an excessive focus on biological aspects and a reproduction of gender inequalities through a focus on risk, personal responsibility, and self-regulation that was moralizing and punitive, as well as focused exclusively on risks in heterosexual relationships [37].

Regarding the period of puberty, transgender participants reported experiencing it with feelings of distress and alarm, particularly at the onset of gender dysphoria and the subsequent concerns associated with body image.

Figure 4 summarizes the main areas of intervention that should be addressed to achieve inclusive sex education. In continuity with what has been proposed in the bio-psycho-social approach, a sex education able to achieve the goal of developing and enhancing a positive attitude to the body and pleasure, as well as a healthy and respectful relationship with self and others, needs to work on several areas, such as: assertiveness (respect for one’s rights and sexual safety), body image, the expression of one’s sexual identity, and knowledge, free from taboos and prejudices.

Indeed, due to a description of body changes that did not propose representations accessible to transgender people, a perception was generated in them that their bodies would inevitably undergo changes, a source of inevitable discomfort [35]. In some parts of the world, so-called “homosexual acts” are prohibited by law resulting in prison sentences and, in some cases, the death penalty. The existence of laws such as these imply on LGBTQIA+ people, regardless of their narrow sphere of daily life, resistance and discomfort in expressing their sexual identity (i.e., the phenomenon called “in the closet”). In younger people, this state can lead to delays and discomfort in knowing, being aware of, and exploring one’s gender, sexuality, and relational experience [38]. People in the LGBTQIA+ community may, in fact, find themselves struggling to learn what it means to form a relationship and having to defend themselves from family members who do not accept their identity [39].

### 3.4. The Importance of the Web Context

In this landscape, LGBTQIA+ people are trying to fill the void of institutionalized sex education by sharing important information about sexual health and rights. Today, social media is a new dissemination medium used by SMs and GMs to achieve this goal [40].

For LGBTQIA+ youths, online access to information gives the opportunity for an equality of access. Online sexual health information has been shown to provide LGBTQIA+ youths with control over their sexual health needs, giving them the opportunity to self-educate on topics not covered in traditional sexuality education, like diverse sexuality, interventions (surgical and non) for transgender individuals, and gender identities [41].

### 3.5. The Importance of Inclusive Dialogue with Key Adults: The Role of Parents, Teachers, and Educators

It seems that promotion of sexual knowledge by the family can be effective in improving sexual safety. It seems, moreover, that family-based educational interventions, in which parent–child communication skills are improved, lead to significant reductions in the enactment of risky sexual behaviors [25].

Parent–child LGBTQIA+ communication about sexuality can be fraught with barriers, as youth often experience anxiety, fear, and discomfort. Young people often avoid initiating conversations about sex after hearing negative comments from their parents about SM and GM people. Moreover, in some cases, parents are believed to lack useful information. Parental support is fundamental in influencing youths’ willingness to engage in conversations regarding sex, making them more receptive and proactive to issues pertaining to sexual health promotion.

Topics such as romantic relationships, emotions, and dating safety may be rare in communication between parents and SM and GM children [42].

An interesting part of studies also focuses on understanding aspects supporting environmental and structural elements as a form of inclusion in schools.

Beasy and collaborators examined teachers’ and staff members’ perceptions, underscoring how the design and arrangement of spaces can either facilitate or hinder inclusion [43]. “Safe spaces” are particularly significant; they are environments where LGBTQIA+ students can express their identities without fear of discrimination or bullying. The study suggests that creating these spaces involves not only physical considerations—such as designated areas for LGBTQIA+ discussions and activities—but also the establishment of supportive relationships among staff and students. This can include training for teachers on LGBTQIA+ issues, promoting awareness and acceptance among the broader student body, and actively engaging LGBTQIA+ students in the process of defining what a safe space means to them.

Furthermore, the research points to the importance of ongoing dialogue among educators about their roles in creating inclusive environments. By negotiating and establishing these safe spaces, teachers and staff can contribute to a more supportive school culture that values diversity and fosters a sense of belonging for all students.

This approach aligns with broader educational goals of equity and inclusion, demonstrating how environmental and structural elements can be leveraged to support marginalized groups within educational settings.

In school settings, professionals involved in adolescent sex education recognize that this environment can be dangerous for SM and GM youth as victims of discrimination [44].

GLSEN (Gay, Lesbian, and Straight Education Network), a U.S.-based educational organization that addresses bullying, harassment, and discrimination based on gender identity, gender expression, and sexual orientation, has found that non-binary students perform worse academically than heterosexual students, but also compared to their lesbian, gay, and bisexual counterparts. According to data reported (2015), feelings of unsafety, fear, and experiences of harassment related to gender expression, discomfort, and gender-related insults are the motivations that lead young people to avoid and/or negative experiences with respect to the school environment.

A key finding to consider is that transphobic comments did not come from peers but, for 63.5% of respondents, from teachers and/or other adult figures in the school setting [44].

Among the adults involved, educators play an important role in this area. Therefore, it is a striking finding that most education professionals indicate that they do not have adequate knowledge to work with SM and GM students [45].

In the meta-analysis on LGBTQ-inclusive sexuality education, O’Farrell et al. (2021) discuss challenges faced by teachers. They suggest that there is an ambivalence and anxiety from some facilitators to carry on inclusive sexuality education that can be explained with their own stigma or perceived inability to talk about these topics because of a lack of information and training.

The literature, in fact, states that skills related to affirming young people’s identities, using neutral and specific language and creating relevance, are not often considered in teachers’ training [41].

Educators require training that broadens their understanding of biological sex, gender identity, sexual and romantic orientation, and sexual behavior.

The recognition that language is constantly evolving and that young people have the right to self-identify should be integrated into the teaching of basic terminology.

Considering that the biopsychosocial approach appears to be the most comprehensive in the implementation of effective sex education interventions, it is believed that the use of the integrated approach (which works on cognitive, emotional, and socio-cultural aspects regarding sexuality) can be a good reference for the training of educators. Through the integrated training methodology, it is possible to propose the study of the most effective models, in parallel with the experimentation of training experiences (through simulations, bioenergetics exercises, and graphic and creative exercises) that allow future sexuality educators to become more aware of implicit and explicit messages, taboos, personal beliefs, and values that they carry in their interventions. In this way, it is possible to imagine that educators can convey messages and content that are more inclusive and able to increase the self-determination resources of the target youth.

A recent study conducted in Europe [46] confirmed the importance of inclusive school policies for LGBTIyouth. Key takeaways include:Improved Safety and Well-Being: Schools with more inclusive policies lead to lower odds of lack of safety and concealment among LGBTQIA+ youth. This indicates that when schools actively promote inclusivity, students feel safer and more accepted.Higher Life Satisfaction: LGBTQIA+ youth in inclusive environments report higher levels of life satisfaction, suggesting that acceptance and support within the school context significantly enhance their overall well-being.Impact of Teacher Training: Training teachers to be more inclusive positively affects school climate. It is associated with reduced feelings of depression and sadness among LGBTQIA+ youth, as well as reduced bias-based violence.Role of Inclusive Curricula: Including LGBTQIA+ topics in the curriculum not only promotes visibility but also correlates with decreased experiences of harassment, both general and bias-based. This highlights the importance of representation in educational materials.Visibility vs. Concealment: Teacher training is linked to greater visibility of LGBTQIA+ youth, which in turn reduces their need to conceal their identities. This suggests that supportive educators can create an environment where students feel comfortable being themselves.

Overall, the study emphasizes that comprehensive strategies involving both teacher training and inclusive curricula are essential for fostering a safe and supportive school environment for LGBTQIA+ youth. By equipping educators with the necessary skills and knowledge, and integrating inclusive content into the curriculum, schools can create an environment where LGBTQIA+ students feel accepted and protected. This dual approach addresses both the immediate need for a supportive atmosphere and the long-term goal of systemic change in attitudes and practices within the educational system.

Moreover, the findings underscore the critical role that inclusivity in teacher training and curricula plays in fostering a safer school environment for LGBTQIA+ youth. By incorporating comprehensive training that emphasizes diversity and the enumeration of protected groups, teachers are better equipped to create a supportive atmosphere that actively reduces feelings of depression and sadness among these students (by 10%). Moreover, these inclusive practices are essential in combating bias-based school violence, highlighting the need for educational institutions to prioritize training that promotes understanding and acceptance.

Increased inclusivity in teacher training and curricula was significantly linked to reduced likelihood of perceived lack of school safety (by 17% and 16%, respectively) and to 18% and 14% lower odds of bias-based school violence, respectively.

The significant reduction in the odds of perceived school safety issues and bias-based violence suggests that when educators are trained to recognize and value diversity, they can effectively model positive behavior and set norms that discourage bullying [47]. This aligns with existing literature that emphasizes the necessity of specialized skills in addressing bias-based victimization, indicating that targeted interventions can lead to substantial improvements in school climate and student well-being.

Overall, these findings advocate for the implementation of inclusive policies and training programs as a means to enhance safety and support for marginalized youth, ultimately contributing to a more equitable educational environment. Often, bias-based victimization prevention efforts require specific knowledge and skills [48].

A recent study aimed to gain a comprehensive understanding of the LGBTQIA+ community’s presence in sexuality education and the ways it is addressed. To achieve this, qualitative research was conducted using a collective case study approach within the context of sexuality education activities at two secondary schools in the province of Almería (Andalusia, Spain). Data collection methods included non-participatory observation, semi-structured interviews, and analysis of documentary sources. The findings indicate a significant prevalence of exclusionary perspectives, the perpetuation of discriminatory beliefs, and a tendency to silence. Furthermore, they demonstrate that the LGBTQIA+ community is treated in a superficial manner. Consequently, sexuality education is not viewed as a space for discussing non-hegemonic identities and experiences, nor are these realities recognized as part of the broader spectrum of sexuality. This perspective contributes to the continuation and legitimization of a stigmatized viewpoint, as non-hegemonic identities and experiences are seen as abnormal, rare, exceptional, and taboo [49].

## 4. Discussion

### 4.1. Influence of SRE in LGBTQIA+ Youths

One of the most important factors that has emerged as influencing SRE in GM and SM youths seems to be their relationship with parents. In fact, communication between parents and LGBTQIA+ adolescents may be characterized, prior to their child’s coming out, by a hetero-cis-normative approach [50]. In this case, parents may have more difficulty, as they may be less aware, educated, and comfortable about same-sex sexual behaviors, or raising a transgender or nonbinary person, as well as being influenced by general and, potentially, negative beliefs and attitudes.

In Australia [51], research has found that LGBTQIA+ youth, unlike their heterosexual counterparts, perceive SRE programs as the least useful source of information. In contrast, they regard online information and social media as the most important sources. This datum seems to be strictly connected to the themes that are discussed in such interventions. Regarding the themes, it emerged from a study that from ages 14 to 15, certain issues receive more consideration, like the use of gender-neutral and inclusive language, and the utilization of gender-neutral terms like “partner” rather than “boyfriend” [52,53,54].

### 4.2. Use of a Neutral and Specific Language to Create an Inclusive Environment

Neutral language and specificity both play important roles in creating inclusive environments, particularly in educational settings.

Neutral language serves as an umbrella that encompasses a diverse range of identities, allowing individuals from various sexual orientations and gender identities to feel represented and included. This approach can foster a sense of belonging among young people who might otherwise feel marginalized or overlooked in traditional curricula. By using language that is inclusive and non-specific, educators can create a welcoming atmosphere that encourages all students to engage with the material without fear of exclusion.

On the other hand, specificity acknowledges and names individual identities, providing visibility to particular groups that may otherwise remain invisible in neutral language. This can help students see themselves in the scenarios presented, facilitating deeper reflection on their own experiences and the decisions they face [15]. When specific identities are highlighted, it can lead to richer discussions and a better understanding of the unique challenges and perspectives that different individuals encounter.

Both approaches have their merits: neutral language promotes inclusivity and broad appeal, while specificity fosters recognition and validation of individual experiences. Striking a balance between the two can enhance educational practices, ensuring that all students feel seen, heard, and valued.

### 4.3. What Topics to Include for Inclusive Sex Education?

A 2014 study by the U.S. Centers for Disease Control and Prevention (CDC) [55] found that 72% of high schools in the United States educate students on how to prevent pregnancy, 60% promote the effectiveness of contraception, and 35% instruct students in condom use.

It would be important for them to be included, in order to address issues regarding consent, safety, assertive communication, and negotiating sexual protections with partners. These aspects are of paramount importance, even more so when considering that young people belonging to SMs and GMs are found to be more at risk for sexual coercion, dating violence, and physical and psychological abuse [56].

Comprehensive sex education that includes discussions on consent, safety, and communication can provide them with the tools to navigate these challenges more effectively.

Teaching students about consent is crucial in fostering a culture of respect and mutual agreement in sexual activities. Ensuring safety ensures that students are better prepared to protect themselves in various contexts. In the context of sexual health, teaching students how to communicate assertively helps them to develop healthy relationship dynamics where both partners can voice their opinions and concerns without fear of judgment or retribution.

This education is essential in creating a safe environment where individuals feel empowered to make informed choices about their bodies and relationships.

The CDC’s findings highlight the progress made in sexual health education in U.S. high schools, but also underscore the gaps that still exist.

### 4.4. Narrative Approach in Educational Programs for an Inclusive Sexuality

Building on Plummer’s (1995) [57] work focused on sexual histories, the importance of adopting a “narrative” perspective in interventions on identity components has been increasingly reinforced in order to capture the dynamic meaning of the dimensions that structure human sexuality. Adopting a narrative perspective in interventions on identity components is crucial for capturing the dynamic and multifaceted nature of human sexuality. This approach allows for a more comprehensive and empathetic understanding of sexual identities, fostering a more inclusive and supportive environment for individuals to explore and express their sexuality. This approach contrasts with static models of identity, which often fail to capture the lived realities of individuals. Plummer’s work highlights the importance of these personal stories in understanding the diversity and fluidity of human sexuality.

Gilbert [58] argues that in order to recognize the rights of people in the LGBTQIA+ community, it is necessary to adopt a theory of sexuality that both tolerates and, at the same time, challenges the relationship one has to the categories used both from outside and within the community [59]. By focusing on the narrative aspects of sexual histories, interventions can also challenge and deconstruct harmful societal norms and stereotypes. This approach encourages a more inclusive understanding of sexuality, recognizing the validity of diverse sexual identities and experiences. It aligns with contemporary views on sexuality, which emphasize the importance of personal agency and the right to define one’s sexual identity free from societal constraints.

The issue of identity has been recognized [32] as a central aspect to be considered in sexuality education programs, including a welcoming and representative view of the different forms it can take, overcoming the conception of a “normative” and standardized pathway. In addition to the importance they represent for people living with commitment to the delicate topic of identity exploration, issues such as coming out, the transition process, asexuality, and nonbinary identities prove to be crucial across the board.

In this regard, it may be significant to reflect on the fact that “inclusion” means “welcoming within”; the perspective that educators need to implement most is to offer reflections, analysis, debate, and critical discussion for all and on all aspects of knowing, exploring, and expressing one’s sexual identity.

It is fundamental to consider that an individual may also belong to multiple SMs or GMs and/or other social groups and categories. Organizing interventions by grouping them into a single category, such as “sexual and gender minority”, can be misleading and ineffective in promoting sexual health [60].

### 4.5. Use of Alternative Sources of Information

Among the tools that can be used in order to tell and see diverse experiences represented, two can be proposed: narratives and comics. Narratives are an opportunity to highlight the significance of diversity and contextual sensitivity when one’s identity and Minority Stress are at stake, in situations where it is health that is being undermined: “Stories work well in representing the need, the difficulty and often the anger of choosing how to live” [61]. Through storytelling, people can articulate their experiences, desires, and challenges, providing a more nuanced view of their sexuality.

As far as LGBTQIA+ adolescents are concerned, an interesting option as an opportunity to read narratives is represented by comic books. Numerous lesbian, bisexual, and queer cartoonists have used this form of expression to popularize the various narratives aimed at this age group of the LGBTQIA+ population. Such representations provide an opportunity to examine how these stories are told, the use of iconographic details aimed at portraying romantic relationships between characters, and how representations of non-hetero-cis-normative relationships provide a positive view of LGBTQIA+ characters [62]. Seeing characters who share similar struggles, triumphs, and everyday experiences can be incredibly validating. This representation helps combat feelings of isolation and fosters a sense of belonging and visibility [63].

Moreover, the combination of visual art and storytelling in comics makes them particularly accessible and engaging. For adolescents, the visual aspect can make complex themes more understandable and relatable. The format often allows for a more immediate emotional connection, as facial expressions, body language, and visual metaphors complement the narrative [64].

The comic book medium encompasses a wide range of genres, from superhero tales to slice-of-life stories, fantasy, science fiction, and romance. This diversity means that there are narratives that can cater to various tastes and interests, all while providing meaningful LGBTQIA+ representation.

Delivering sexuality education focused on traditional heterosexuality (i.e., vaginal intercourse and reproduction) is not congruent with the daily experiences of youths today, many of whom are sexually fluid (and/or engage in a wide range of sexual practices regardless of sexual identity [54,65].

## 5. Conclusions

In light of the discussed contributions, it is of paramount importance that sex educators, but also all those who provide training to adolescents (teachers, parents, sport workers, and health professionals) engage in fully implementing the fundamentals of holistic sex education, which, as mentioned, involves an inclusive and cross-cutting epistemology [66,67,68,69,70,71,72,73,74,75,76,77].

Inclusive education should discuss of all different experiences and practices, offering a non-judgmental environment and the possibility of feeling included in the education. In order to do this, it is important to have both a formal and an informal sexual education, suggesting also possible useful websites and/or apps that could be useful [3,78]. Narratives and comics could be two important tools to use with this specific target [54,63].

Different studies have demonstrated diminished well-being for lesbian, gay, bisexual, transgender, queer, intersex, and asexual (LGBTQIA+) youths relative to their peers, with the most results focused on negative experiences at school as key factors associated with these differences [79,80,81,82].

All of this is central not only to make interventions more effective and more responsive to the needs of target audiences, but also to embrace the call of the technical documents to make sex education a process of growth, respect, and acceptance of all and for all.

In this regard, it is deemed necessary to further point out that knowing the specific educational needs of the LGBTQIA+ adolescent population does not mean working exclusively in the creation of ad hoc interventions.

Sexuality education that does not explicitly address diversity is ineffective for young people growing up in a world where sexual and gender diversity is prevalent and understandings of sexuality and gender differ significantly from previous generations. It is important that all young people—regardless of their gender identity—have access to appropriate and accurate information about sexuality that goes beyond the dominant heterosexist approach to include sexual and gender diversity. An inclusive approach to sexuality education should, at a minimum, mean that young people are well prepared for future consensual sexual relationships, whatever that may entail. Through an inclusive approach, all young people will have a better understanding of diversity.

The first limitation is that this is a narrative review, so a systematic one could provide data that are more detailed. Another limitation of the review is that it is based upon a compilation of peer-reviewed English and Spanish language literature. This literature largely comes from Europe and North America, much less often from Latin America and Australia, and Asian or African sources are completely absent. The studies referenced thus approach the subject of investigation from a predominantly Western perspective. The review provides little information on both the online sexual activities in non-Western populations and the non-English academic discourses on these activities. The last limitation that we think it is useful to address is that maybe different reviews could specifically concentrate on the different sexual and gender identities, in order to provide more detailed info on each of them.

What is proposed, instead, as a further perspective of development, is to broaden the attention and sensitivity in writing projects, defining themes, selecting tools and activities-stimulus, in order to make whole groups work, with all the different subjectivities and identities that compose and structure them. As highlighted by others [83,84] inclusivity in the wider school environment is central to LGBTQIA+ students’ sense of safety and belongingness in the school context.

## Figures and Tables

**Figure 1 children-11-00966-f001:**
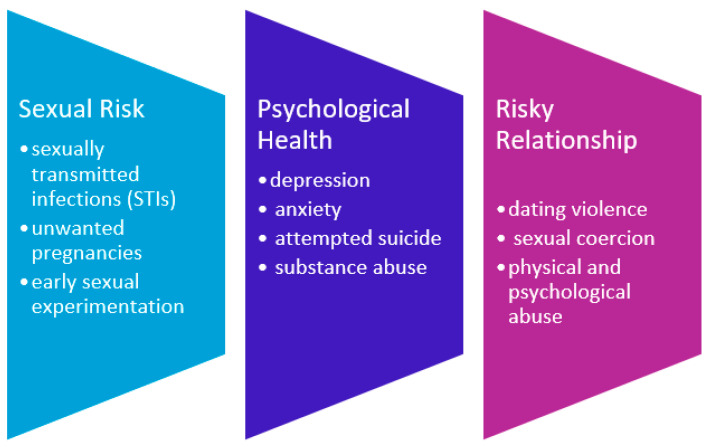
Effects of hetero-cis-normativity sex education.

**Figure 2 children-11-00966-f002:**
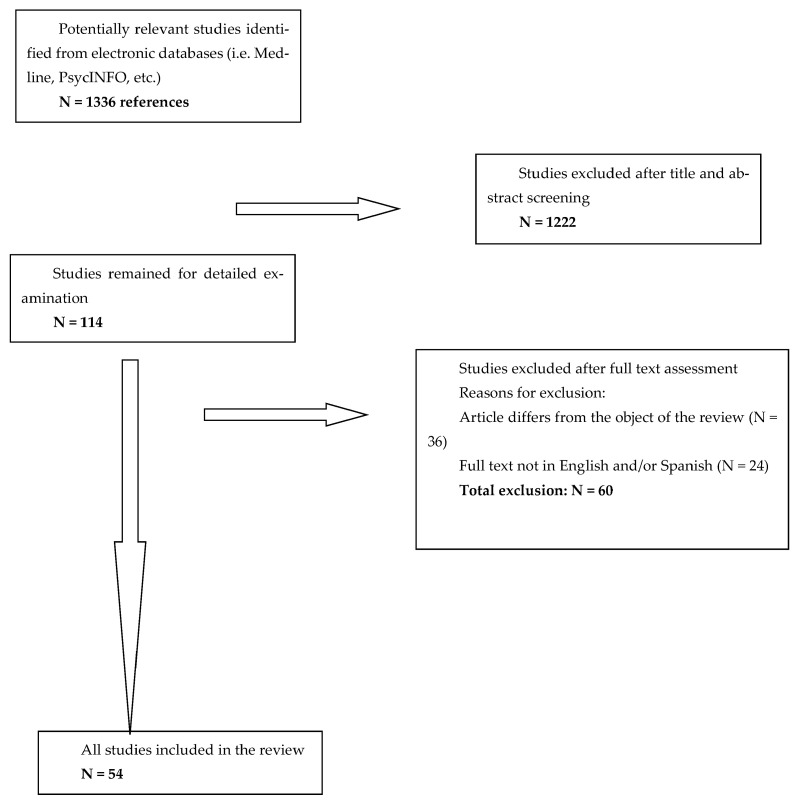
Flow chart of study selection procedure.

**Figure 3 children-11-00966-f003:**
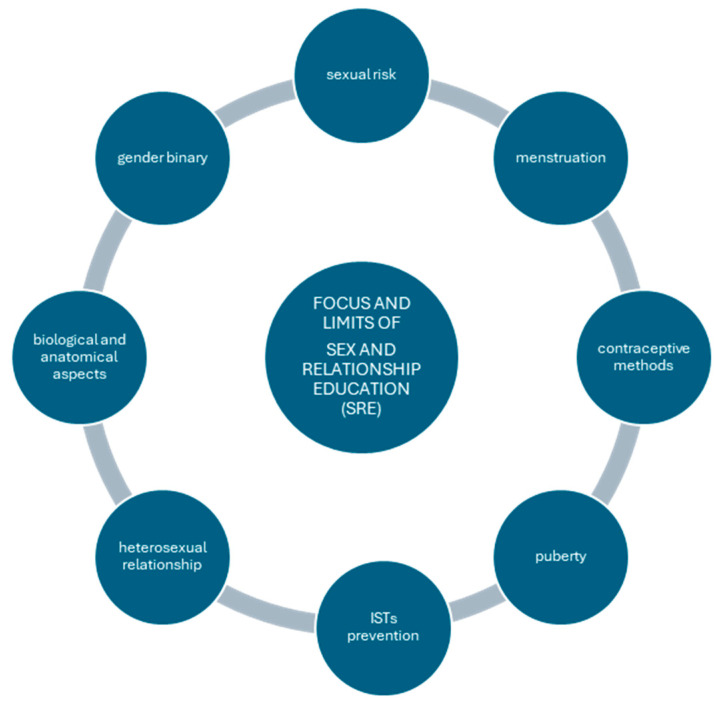
Focus and limits of sex and relationship education.

**Figure 4 children-11-00966-f004:**
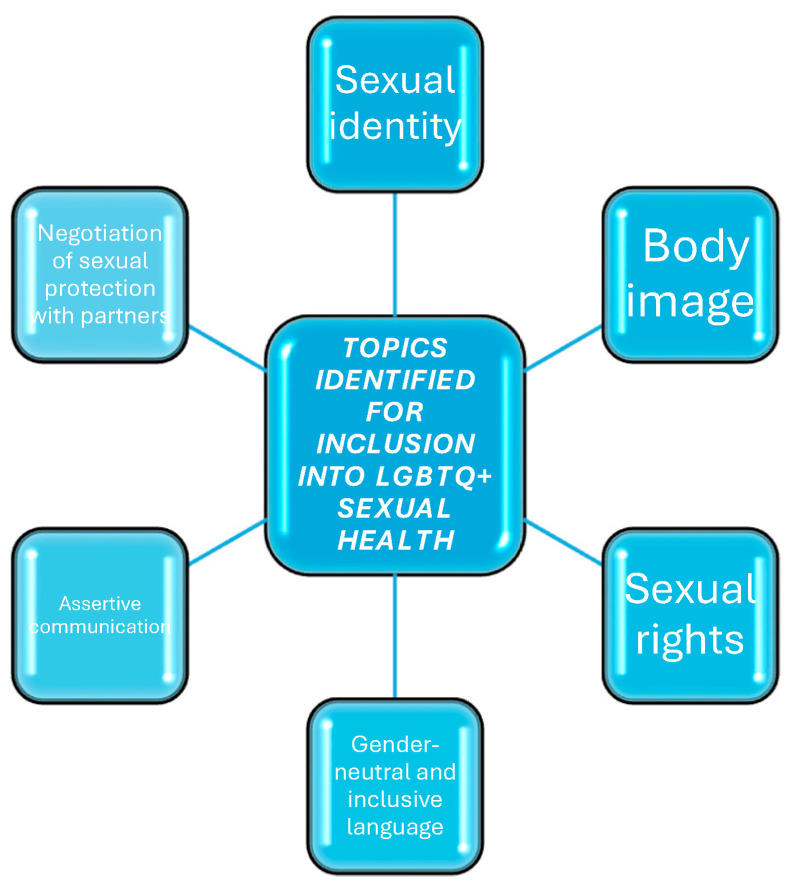
Perspectives and educational needs of LGBTQ+ youth.

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
