# Peer review of "Inclusion Goals: What Sex Education for LGBTQIA+ Adolescents?"

_children, 2024, doi:10.3390/children11080966_

Round 1

Reviewer 1 Report

Comments and Suggestions for Authors

Thank you very much for allowing me to review this very interesting article. You then pointed out some areas for improvement.

In the methodology it is not clear how this review has been carried out, i.e. what type of review it is, how many articles have been reviewed, has a PRISMA been used, what are the inclusion criteria, has critical literature such as CASPE been used, etc. In addition, It should be revised the methodology and the results to make them suitable by adding a summary table as in the PRISMA checklist and add the figure of the P

Comments on the Quality of English Language

I do not have comments about it. It is correct.

Author Response

Reviewer #1: Thank you very much for allowing me to review this very interesting article. You then pointed out some areas for improvement.

In the methodology it is not clear how this review has been carried out, i.e. what type of review it is, how many articles have been reviewed, has a PRISMA been used, what are the inclusion criteria, has critical literature such as CASPE been used, etc. In addition, It should be revised the methodology and the results to make them suitable by adding a summary table as in the PRISMA checklist and add the figure of the P

Reply: Thanks for your comment; we hope the methodology is clearer than before (please see row 1 and rows 11-22 of the “Materials and method” section, pagg. 4-5). We wrote that it is a narrative review and we also added a figure with a flowchart of the selection procedure. Since it is a narrative review we did not added a summary table like in the PRISMA checklist.

Reviewer 2 Report

Comments and Suggestions for Authors

Thank you for this interesting paper reviewing the current literature about sex education for adolescent LGBTQIA+.

The paper is generally well written, though there are some minor grammatical errors that require some attention (probably acquired somewhere in the translation process). There are also some issues around content that need work. 

I have some comments that I hope are helpful. 

1. INTRODUCTION

This section provides a useful background to the topic and highlights the importance of a broad-based (holistic) approach to sex education. It also provides some vital detail about the repercussions of sex education that does not address the experience of all adolescents. 

It would be helpful to provide some definitions here, for example, ‘hetero-cisnormative’, and especially ‘LGBTQIA+’. With the latter, there’s a real danger that all non-hetero sexualities are clumped together, whereas each one separately has its own cluster of novel issues, priorities, and challenges.

Also, there is nothing in the text (here, or elsewhere) that explains the figures (1-3). This should be added. 

Finally, I know the topic is addressed briefly later in the paper, but there should be some consideration in the introduction about the social, cultural, and political aspects (especially the latter in the US) of non-hetero sexualities. Given the current state of the ‘culture wars’, this whole field is now heavily politicised and (in some places) significantly impacts on the delivery of sex education in schools. This adds further complications to an already complex topic (adolescent sexuality). 

2. MATERIALS AND METHODS

You explored all the suitable databases, though it would be helpful to know more about how you selected ‘the most relevant studies’ (line 120). 

3. RESULTS

This section provides much detail and includes some helpful information. I would recommend adding some subheadings and creating a little more structure. Also, there are two figures with no in-text explanations. 

I was pleased to see reference to the creation of safe spaces, and the need for much more information about relationships within sex education. Also, the risk of insensitive and/or irrelevant sex education leading to mental health problems and stigmatisation is a crucial factor, and there are some useful citations here to support that. 

What I didn’t see was any information about how the educators themselves can be trained and prepared to deliver truly holistic sex education. This would be interesting (with examples of successful innovations) if the data exist. I did note the benefits of peer support was highlighted, which was useful. 

4. DISCUSSION

You draw on your key findings and refer to further studies to broaden your discussion. As in the Results section, this would benefit from a clearer structure. 

There are some valid points here, for example the use of alternative sources of information (e.g., comic books) and (more broadly) how a narrative approach can be helpful. But I did find this section (as in the Discussion section) a little unfocused. It would benefit from a deeper consideration of some of the core issues which carefully avoids coalescing sexualities that must be considered separately.

5. CONCLUSION

You make a strong case for a broader approach to sex education, which is excellent. However, I’d like to see more recommendations about how this can be done in practice, which you do start to consider in the final paragraph. But how will this really be operationalised? How can local considerations (social, cultural, political) be taken into account? Does the literature provide examples of good practice?

Finally, you should include any limitations of your review in this section. 

REVIEWER RECOMMENDATIONS 

1.     Add definitions of the core terms and acronyms in the paper.

2.     Add text to explain the Figures.

3.     In the introduction, consider briefly the social, cultural, and political factors that may impact on holistic sex education. 

4.     In the Materials and Methods section, add more detail about how you selected the ‘most relevant’ studies. 

5.     Add more structure to the Results and Discussion sections.

6.     If the data exist, provide examples of how educators can be better prepared to deliver appropriate material (if the data don’t exist, then state this). 

7.     In the conclusion, add any limitations for this review, and provide clearer recommendations of good practice and/or how holistic sex education can be delivered. 

Comments on the Quality of English Language

The English is generally good, though there are some anomalies (e.g., line 348, 'coated') that need correcting. 

Author Response

Reviewer #2: Thank you for this interesting paper reviewing the current literature about sex education for adolescent LGBTQIA+.

The paper is generally well written, though there are some minor grammatical errors that require some attention (probably acquired somewhere in the translation process).

Reply: Thanks for your comment; you can see that in the paper we checked and changed different sentences that now we think are better from the linguistic point of view (i.e. rows 11-16 at pag 9 and rows 2-6 at pag 10)-

There are also some issues around content that need work. 

I have some comments that I hope are helpful. 

  1. INTRODUCTION

This section provides a useful background to the topic and highlights the importance of a broad-based (holistic) approach to sex education. It also provides some vital detail about the repercussions of sex education that does not address the experience of all adolescents. 

It would be helpful to provide some definitions here, for example, ‘hetero-cisnormative’, and especially ‘LGBTQIA+’. With the latter, there’s a real danger that all non-hetero sexualities are clumped together, whereas each one separately has its own cluster of novel issues, priorities, and challenges.

Reply: Thanks for your comment. We expanded the definitions of “hetero-cisnomative” and “LGBTQIA+” (please see rows 18-30, pag. 2 in the introduction section).

Also, there is nothing in the text (here, or elsewhere) that explains the figures (1-3). This should be added. 

Reply: Thanks again. Figure 1 has been introduced and explained at rows 41 and 45-50, pag. 2 in the introduction section. Figure 2 (now Figure 3) has been discussed at rows 36 and 43-38, pag. 6 in Results section. Figure 3 (now Figure 4) has been introduced and explained at rows 1-7, pag. 8 in the results section)

Finally, I know the topic is addressed briefly later in the paper, but there should be some consideration in the introduction about the social, cultural, and political aspects (especially the latter in the US) of non-hetero sexualities. Given the current state of the ‘culture wars’, this whole field is now heavily politicised and (in some places) significantly impacts on the delivery of sex education in schools. This adds further complications to an already complex topic (adolescent sexuality). 

Reply: Thanks for your comment. We expanded the discussion on the social, cultural and  political aspects aspect in the introduction and the discussion (please see rows 7-17, pag. 4 in the introduction section).

  1. MATERIALS AND METHODS

You explored all the suitable databases, though it would be helpful to know more about how you selected ‘the most relevant studies’ (line 120). 

Reply: Thanks for your comment; we hope the methodology is clearer than before (please see row 1 and rows 11-22 of the “Materials and method” section, pp. 4-5). We wrote that it is a narrative review and we also added a figure with a flowchart of the selection procedure.

  1. RESULTS

This section provides much detail and includes some helpful information. I would recommend adding some subheadings and creating a little more structure. Also, there are two figures with no in-text explanations. 

Reply: We provided subsections to make the text more agile to read (pp 6-8)

I was pleased to see reference to the creation of safe spaces, and the need for much more information about relationships within sex education. Also, the risk of insensitive and/or irrelevant sex education leading to mental health problems and stigmatisation is a crucial factor, and there are some useful citations here to support that. 

What I didn’t see was any information about how the educators themselves can be trained and prepared to deliver truly holistic sex education. This would be interesting (with examples of successful innovations) if the data exist. I did note the benefits of peer support was highlighted, which was useful. 

Reply: Thanks for your comment; we added a paragraph about it (please see rows 7-17 of the “Results” section, p. 10).

  1. DISCUSSION

You draw on your key findings and refer to further studies to broaden your discussion. As in the Results section, this would benefit from a clearer structure. 

There are some valid points here, for example the use of alternative sources of information (e.g., comic books) and (more broadly) how a narrative approach can be helpful. But I did find this section (as in the Discussion section) a little unfocused. It would benefit from a deeper consideration of some of the core issues which carefully avoids coalescing sexualities that must be considered separately.

Reply: Thanks for your comments. We added some paragraphs with other surces and provided subsections to make the text more agile to read (pp 12-14)

  1. CONCLUSION

You make a strong case for a broader approach to sex education, which is excellent. However, I’d like to see more recommendations about how this can be done in practice, which you do start to consider in the final paragraph. But how will this really be operationalised? How can local considerations (social, cultural, political) be taken into account? Does the literature provide examples of good practice?

Reply: Thanks for your comment; we added a paragraph about it (please see rows 5-9 of the “Conclusions” section, p. 14, where we included also a new reference showing a good practice of apps for LGBTQIA+ online sexual education).

Finally, you should include any limitations of your review in this section. 

Reply: Limitations have been included (please see rows 10-20 of the “Conclusions” section, p. 14)

REVIEWER RECOMMENDATIONS 

  1. Add definitions of the core terms and acronyms in the paper.
  2. Add text to explain the Figures.
  3. In the introduction, consider briefly the social, cultural, and political factors that may impact on holistic sex education. 
  4. In the Materials and Methods section, add more detail about how you selected the ‘most relevant’ studies. 
  5. Add more structure to the Results and Discussion sections.
  6. If the data exist, provide examples of how educators can be better prepared to deliver appropriate material (if the data don’t exist, then state this). 
  7. In the conclusion, add any limitations for this review, and provide clearer recommendations of good practice and/or how holistic sex education can be delivered. 

Reply: Thanks again for all the comments. We think we replied to all of them in the previous sections

Round 2

Reviewer 1 Report

Comments and Suggestions for Authors

Thank you they have improved.

Reviewer 2 Report

Comments and Suggestions for Authors

Thank you for this revised version of your paper. I can see that you have addressed my recommendations, and I have no further comments to make.